# The Role of Cytokines in the Metastasis of Solid Tumors to the Spine: Systematic Review

**DOI:** 10.3390/ijms24043785

**Published:** 2023-02-14

**Authors:** Wojciech Łabędź, Anna Przybyla, Agnieszka Zimna, Mikołaj Dąbrowski, Łukasz Kubaszewski

**Affiliations:** 1Adult Spine Orthopaedics Department, Poznan University of Medical Sciences, 61-545 Poznan, Poland; 2Department of Cancer Immunology, Poznan University of Medical Sciences, 60-806 Poznan, Poland; 3Institute of Human Genetics, Polish Academy of Sciences, 60-479 Poznan, Poland

**Keywords:** cytokine, cancer, metastases, spine, bone

## Abstract

Although many studies have investigated the role of cytokines in bone metastases, our knowledge of their function in spine metastasis is limited. Therefore, we performed a systematic review to map the available evidence on the involvement of cytokines in spine metastasis in solid tumors. A PubMed search identified 211 articles demonstrating a functional link between cytokines/cytokine receptors and bone metastases, including six articles confirming the role of cytokines/cytokine receptors in spine metastases. A total of 68 cytokines/cytokine receptors were identified to mediate bone metastases; 9 (mostly chemokines) played a role in spine metastases: CXC motif chemokine ligand (CXCL) 5, CXCL12, CXC motif chemokine receptor (CXCR) 4, CXCR6, interleukin (IL) 10 in prostate cancer, CX3C motif chemokine ligand (CX3CL) 1 and CX3C motif chemokine receptor (CX3CR) 1 in liver cancer, CC motif chemokine ligand (CCL) 2 in breast cancer, and transforming growth factor (TGF) β in skin cancer. Except for CXCR6, all cytokines/cytokine receptors were shown to operate in the spine, with CX3CL1, CX3CR1, IL10, CCL2, CXCL12, and CXCR4 mediating bone marrow colonization, CXCL5 and TGFβ promoting tumor cell proliferation, and TGFβ additionally driving bone remodeling. The number of cytokines/cytokine receptors confirmed to mediate spinal metastasis is low compared with the vast spectrum of cytokines/cytokine receptors participating in other parts of the skeleton. Therefore, further research is needed, including validation of the role of cytokines mediating metastases to other bones, to precisely address the unmet clinical need associated with spine metastases.

## 1. Introduction

Cancer metastasis is a complex process that involves access to the systemic circulation, cell migration, loss of intercellular cohesion, angiogenesis, and evasion of local immune responses. Bone metastases are common in cancer and typically predict poor prognosis. There are certain cancers with a high prevalence of bone metastases, namely prostate (85%), breast (70%), lung and kidney (40%, both) [1]. The most common site of bone metastasis is the axial skeleton, including the spine (87%), ribs (77%), pelvis (63%), and the proximal humeri and femora (53%) [1]. In several types of cancer, including prostate, breast and lung cancer, bone metastases are associated with markedly shorter survival [2]. In the case of spinal metastases, only 10–20% of patients survive more than two years from diagnosis [3]. Up to one-third of cancer patients will develop spine metastases; in the US alone, approximately 180,000 cases with spine metastasis are diagnosed annually [3]. Severe pain refractory to treatment is the most common clinical symptom associated with spine metastases. Furthermore, bone metastases lead to significant morbidity, including pathological fractures, the need for radiotherapy to reduce pain or bone remodeling, the need for surgery to repair or prevent bone damage, spinal cord compression, and hypercalcemia [4]. These skeletal-related events (SRE) are associated with worse mobility, reduced social functioning, and a lower quality of life. The treatment of spine metastases is mostly palliative, and it is aimed at preserving or improving quality of life, as well as the protection or restoration of neurological functions and spinal stability [3]. Currently, bone-targeted agents are the mainstay of treatment of spine metastases, and their use can limit the incidence of SREs; however, their influence on survival is rather limited [5].

Cancer cell diffusion through the venous system, mostly via Batson’s venous plexus, is the major route of spinal metastasis; more rarely, dissemination occurs via the arterial system and by contiguity [6]. Preferential seeding of some types of cancer to the spine appears to be a consequence of a dense vasculature in the red bone marrow resulting in a lower blood flow through this tissue. However, various signaling and adhesion molecules that are highly abundant in the spine tissue cannot be excluded as principal factors contributing to tumor cell invasion and growth in some types of cancer.

Secondary bone cancers can be divided into osteolytic and osteoblastic lesions [7,8]. Osteolytic lesions, characterized by intensive bone resorption, are a more common form of metastasis, and they are especially frequent in breast, prostate and renal cancer. Osteoblastic lesions, often observed in prostate cancer, are associated with abnormal bone formation, which is structurally weaker and more prone to fractures. Nevertheless, many lesions involve both bone formation and resorption processes; these mixed lesions are commonly found in breast cancer.

The preferential spread of cancer to some organs has been elegantly illustrated by the seed and soil hypothesis formulated by Stephen Paget [9]. Paget’s hypothesis suggesting the critical role of a microenvironment in some organs for metastasis has been confirmed by several studies. We know now that tumor cells metastasize only into favorable environments rich in growth factors and cytokines that can enhance homing and growth. Recently, the seed and soil concept has been further developed to include the pre-metastatic niche model that assumes that primary tumors secrete factors that prepare the target organs for colonization. According to this expanded theorem, the molecular priming of tumor cells by stroma at the primary site, combined with the pre-conditioning of the secondary organ, is necessary to facilitate a metastatic spread. In bone metastases, homing of cancer cells is thought to be regulated by crosstalk between factors that are produced by the bones and their receptors expressed on the tumor cells’ surface.

Following endothelial transmigration and the dissemination of cancer cells into the blood system, the multi-step process of metastasis to the bone involves (i) the colonization of the bone marrow by circulating tumor cells, followed by (ii) a dormancy period when cancer cells enter a quiescent state after adapting to the bone marrow microenvironment, (iii) the reactivation of dormant cells and subsequent proliferation, and (iv) reconstruction of the bone tissue by cancer cells [10]. A vast number of cytokines and cytokine receptors mediate these processes by facilitating crosstalk between cancer and bone cells. For example, motif chemokine ligand (CXCL) 12 (also known as SDF-1) and its receptor CXC motif chemokine receptor (CXCR) 4 coordinate the adhesion of circulating cancer cells to bone marrow endothelial cells and their migration across the vessel wall [11]. Bone morphogenetic protein 7 and transforming growth factor (TGF) β2 present in bone marrow were shown to promote dormancy; the subsequent inhibition of their respective signaling pathways is required for the reactivation of cancer cells. The subsequent recruitment of osteoclasts by tumor cells is critical for metastatic bone remodeling. The secretion of Parathyroid hormone-related protein (PTHrP) from tumor cells induces the production of receptor activator of nuclear factor-κB ligand (RANKL) by osteoblasts [12]. The interaction between RANKL on osteoblast surfaces and its receptor RANK on osteoclast precursors initiates osteoclastogenesis. In addition to PTHrP, tumor-derived interleukin (IL) 6 also acts on osteoblasts to stimulate RANKL production. These indirect, RANKL-mediated mechanisms act in parallel with the direct stimulation of osteoclast formation by tumor necrosis factor (TNF) α secreted from tumor cells. Additionally, IL8 has been shown to stimulate osteoclastogenesis indirectly by increasing RANKL on osteoblasts and directly by interacting with CXCR1 on osteoclast surfaces [13]. In addition, IL11 released from bone marrow stromal cells induced by tumor cells can stimulate osteoclast differentiation and activity [14]. Osteoclasts’ activation induces bone resorption, resulting in the release of TGFβ1 deposited in the bone extracellular matrix, which in turn promotes the proliferation of tumor cells and thus feeds forward the cycle of tumor growth and bone remodeling [15]. Although a large body of evidence has confirmed the role of several cytokines in bone metastases, only a very limited number of cytokine-targeting drugs have been shown in clinical trials to downregulate bone metastases. The most known example is denosumab, an anti-RANKL antibody, which was approved by The United States Food and Drug Administration in 2011 to treat bone loss in patients with prostate or breast cancer undergoing hormone ablation therapy. In addition to its positive effect on bone loss, denosumab was shown to reduce the incidence of new vertebral fractures in patients with non-metastatic breast and prostate cancer [16,17].

Therefore, a vast number of cytokines appear to mediate each step of the bone metastasis process, from colonization to bone tissue remodeling. Although a large number of studies investigated the role of cytokines in bone metastases, our knowledge of their function in spine metastasis is limited. In particular, the current literature lacks a complex overview of the involvement of cytokines across the four major steps of metastasis to the spine outlined above. Therefore, we performed a systematic review to map the available evidence on the role of cytokines in the metastasis of solid tumors to the spine and to systematize their involvement in the cancer cell invasion, dormancy, proliferation, and remodeling of spine tissue.

## 2. Methods

The present review was performed in accordance with the Preferred Reporting Items for Systematic Reviews and Meta-Analyses (PRISMA) 2020 statement: an updated guideline for reporting systematic reviews [18]. The protocol of this systematic review has been prospectively registered at the Open Science Network (OSF, DOI 10.17605/OSF.IO/R5UFW).

### 2.1. Search Strategy

We systematically searched all publications from PubMed until 30 April 2022 without any restriction to language. The terms “bone metastases” and “cytokines” were used. Abstracts identified by the search were screened, and articles found to be relevant to the topic of interest were shortlisted. All articles except reviews and case reports were considered. Full-length papers demonstrating a functional link between cytokine and bone metastases and spine metastases, in particular, were included in the final systematic review. WŁ and ŁK independently reviewed abstracts and full-text articles. Disagreements were resolved by discussion between the two reviewers.

The database search yielded 2413 records, including 1903 original research articles (Figure 1). Overall, 1684 articles were excluded after reviewing the title and abstract, and the full text of the remaining 349 articles was evaluated for final inclusion. A total of 221 articles demonstrated a functional link between cytokines/cytokine receptors and bone metastases, including six that confirmed the role of cytokines/cytokine receptors in metastasis to the spine.

### 2.2. Data Items

The extracted data included the name of the first author, publication year, type of primary tumor, name of the cytokine or cytokine receptor, study setting and findings.

## 3. Results and Discussion

In total, 68 cytokines/cytokine receptors were identified to play a role in bone metastases in 12 types of cancer, most often in breast and prostate cancers (48 and 23 cytokines/cytokine receptors, respectively, Appendix A). However, only nine cytokines (mostly chemokines)/cytokine receptors (in four types of cancer) demonstrated a functional role in spine metastases, including CXCL5, CXCL12, CXCR4, CXCR6, and IL10 in prostate cancer, CX3C motif chemokine ligand (CX3CL) 1 and CX3C motif chemokine receptor (CX3CR) 1 in liver cancer, CC motif chemokine ligand (CCL) 2 in breast cancer, and TGFβ in skin cancer (Table 1, Figure 2).

### 3.1. CX3CL1 and CX3CR1

Hepatocellular carcinoma (HCC) patients have an increased expression of chemokine CX3CL1 and its receptor CX3CR1 in spinal metastases [19]. An analysis of the interaction between these proteins revealed that bone marrow endothelial cells (BMEC) secrete CX3CL1 to enhance metastasis via interaction with CX3CR1 on the cancer cell surface (see Figure 2 for a graphical summary). The overexpression of CX3CR1 in HCC cells was shown to promote spinal metastases in nude mice. Furthermore, concomitant injection of HCC mixed with BMEC with knocked-down CX3CL1 reduced the size of bone tumors [19].

CX3CR1 is also expressed on breast cancer cell surfaces [25]. The dissemination of human breast cancer cells to the bone is lower in CX3CL1^−/−^ mice. Conversely, overexpression of CX3CR1 in human breast cancer cells increases homing to the bone by mediating both adhesion to the endothelium and extravasation. Moreover, serum samples from patients with spinal metastases from the lung, kidney, and prostate contained significantly higher levels of CX3CL1 than in healthy controls, which suggests the potential involvement of CX3CL1 in these types of cancer [26]. Studies on hepatocellular carcinoma spinal metastasis revealed that the function of BMEC-derived CX3CL1 is regulated by proteolytic cleavage by ADAM17. CX3CL1 was shown to act via the intercellular adhesion molecule 1 in BMECs to facilitate the metastatic extravasation of non-small-cell lung cancer cells in the spine.

### 3.2. IL10

IL10 was shown to inhibit the growth of primary prostate tumors and metastases in prostate cancer [24,27]. The injection of human prostatic adenocarcinoma cell line PC-3 into severe combined immunodeficient mice (SCID) mice results in bone metastases, approximately 70% of which locate to the vertebrae. The overexpression of IL10 in PC-3 cells or IL10 treatment reduces the number of metastases to the spine (see Figure 2), while in the metastases that did occur, IL10 expression was frequently lost.

### 3.3. CCL2

The expression of the chemokine CCL2 was shown to be reduced in a highly metastatic variant of murine mouse breast cancer 4T1E cells [22]. Overexpression of CCL2 in a highly metastatic variant of 4T1E cells injected into mice reduces metastatic burden in the spine (see Figure 2). Conversely, CCL2 silencing in less metastatic parental cells increases metastases to the spine. Interestingly, the manipulation of CCL2 expression in that model affected metastases to other organs to a much lower extent [22]. However, most of the evidence suggests a pro-metastatic function of CCL2. Increased levels of CCL2 in prostate, breast, gastric, and other types of cancer are associated with more advanced disease and an unfavorable prognosis. In prostate cancer, CCL2 originating from tumor cells and stroma induce the differentiation of osteoclasts from their precursors, their activation and subsequent bone resorption [28]. In bone metastases of breast and prostate cancer, CCL2 expression is driven by PTHrP and results in bone colonization by cancer cells and osteolysis [29]. Additionally, the interaction of CCL2 with CCR2 on prostate cancer cells drives the release of vascular endothelial growth factor (VEGF) A, which, in turn, stimulates angiogenesis and tumor growth. CCL2 knockdown in human prostate adenocarcinoma cell line PC-3 diminishes tumor growth in the bone of SCID mice [30]. Similarly, anti-CCL2 antibody reduces metastatic burden in the bone of mice injected with human prostate cancer cells and decreases bone resorption and microvessel density [31].

### 3.4. CXCR6, CXCR4, and CXCL12

CXCR6 ligand CXCL16 facilitates mesenchymal stem cell recruitment into prostate tumors and their subsequent conversion into cancer-associated fibroblasts [21]. In turn, these cells produce CXCL12, which binds to CXCR4 on tumor cells, inducing epithelial to mesenchymal transition (EMT) and thus facilitating metastasis to the bone (Figure 2). The inhibition of CXCR4 in the murine prostate carcinoma cell line RM1 injected into mice resulted in a decreased number of disseminated tumor cells in the spine and other bone locations. Conversely, stimulation of EMT by CXCL12 in RM1 cells promoted spine metastasis in wild-type but not in CXCR6^−/−^ animals. The role of the CXCL12/CXCR4 system in bone metastasis in prostate cancer has been confirmed in several in vivo studies utilizing specific inhibitors and transgenic mice [11,32,33,34,35,36,37]. Dai et al. demonstrated that inhibition of CXCL12 reduces the number of bone metastases [34]. Interestingly, inhibition of the CXCL12/CXCR4 axis through plerixafor prevented the initial establishment of bone metastases without an impact on the growth of the already established secondary bone tumors [35]. Using several human prostate cancer cell lines, Wang et al. mapped the location of metastatic sites in athymic mice and found that prostate cancer cells preferentially seed to osteoblast-rich bone surfaces in a CXCL12/CXCR4-dependent manner [36].

Moreover, the interaction between CXCL12 and CXCR4 seems to be crucial not only for the homing of prostate cancer cells to the bone but also for overcoming tumor resistance to chemotherapy. Inhibition of CXCR4 by plerixafor sensitized metastatic bone tumors to docetaxel could have a profound impact on clinical practice since all metastatic castration-resistant prostate cancer patients eventually develop resistance to taxanes [37]. Therefore, targeting the CXCR6/CXCL12/CXCR4 pathway could be a viable strategy in prostate cancer patients with metastases in the spine and other bone locations. The expression of CXCR4 in primary breast cancer tumors in patients with locally advanced disease is associated with increased metastases and rapid tumor progression [38]. The inhibition of CXCR4 does not affect the growth of primary tumors in mouse models of triple-negative breast cancer (TNBC) but limits the number of bone metastases, which was further improved by the addition of chemotherapy [38]. Price and colleagues found that dormant breast cancer cells mostly occupy perisinusoidal vascular regions in bone marrow rich in CXCL12, while intensively proliferating cells were found in nonsinusoidal regions [39]. Interestingly, the inhibition of CXCL12 results in the release of dormant breast cancer cells from bone marrow to peripheral circulation. Given the protective role of the bone marrow environment against chemotherapy, which appears to play a role in late relapses in breast cancer, the authors of this study speculated that targeting CXCL12 could be utilized to relocate dormant cancer cells from the bone to the circulation, where they could be more sensitive to chemotherapy. This hypothesis has been confirmed by Xiang et al., who reported that the addition of CXCR4 antagonist POL5551 to chemotherapy was more effective in reducing bone tumor burden than chemotherapy alone in a mouse model of TNBC [38]. Furthermore, the overexpression of CXCR4 in neuroblastoma cells was shown to increase the number of bone marrow metastases [40]. The interplay between CXCR4 and CXCR7 determines organ preference during metastasis. While CXCR7 overexpression resulted in more frequent metastasis to the adrenal glands and the liver, and CXCR4 promoted dissemination to the liver and the lungs, the overexpression of both receptors appears necessary for the selective homing of neuroblastoma cells to the bone [41]. Finally, a reduced bone tumor burden was demonstrated in CXCR4^−/−^ mice injected with human small-cell lung cancer cells [42].

### 3.5. CXCL5

Roca et al. [20] demonstrated the role of the proinflammatory chemokine CXCL5 in murine prostate cancer cell growth in the bone using vertebral vesicles implanted into mice. They found that the induction of cancer cell apoptosis results in the increased infiltration of inflammatory cells and accelerated tumor growth. These effects were attenuated in CXCL5^−/−^ mice, indicating that CXCL5 mediates the expansion of prostate cancer cells in the spine. Mechanistically, the efferocytosis of prostate cancer cells by macrophages induces expression of CXCL5, resulting in inflammation and immunosuppression, which stimulates tumor growth (see Figure 2 and Figure 3). Furthermore, these authors reported increased CXCL5 serum levels in prostate cancer patients with metastatic disease compared to those with less advanced cancer or healthy controls. CXCL5 and its receptor CXCR2 were also reported to induce the proliferation of breast cancer cells and bone colonization ex vivo [43,44]. Moreover, CXCL5 appears to restart proliferation in quiescent breast cancer cells, which indicates its potential role in the reactivation of dormant cancer cells that disseminate to the bone [43,44].

### 3.6. TGFβ

TGFβ operating in the vicious cycle of bone destruction and cancer cell proliferation acts on breast cancer cells to secrete PTHrP, which subsequently induces the release of RANKL from osteoblasts. In turn, RANKL activates osteoclasts, resulting in bone resorption, which causes the activation of latent TGFβ deposited in the bone matrix. This newly activated TGFβ stimulates cancer cells in a feed-forward mechanism, resulting in the amplification of bone tissue remodeling ([15,47], see Figure 2 and Figure 3). Inhibition of TGFβ receptor (TGFBR) 1 reduces metastasis incidence in the lung, spine and other bones in mice injected with a highly metastatic variant of human melanoma cell line MDA-MB-435 (formerly described as a breast cancer cell line) [23]. TGFβ function in bone remodeling by melanoma cells is at least partially controlled by mothers against decapentaplegic homolog (SMAD) 7, which acts via a negative feedback loop in the intracellular TGFβ signaling cascade. Cardiac inoculation of human melanoma cells 1205Lu overexpressing SMAD7 reduces osteolysis and improves survival [48].

TGFβ has also been implicated in bone metastases in other types of cancer, including breast, prostate, kidney, and lung. The elevated expression of TGFBR1 and the activation of TGFβ signaling has been observed at the tumor–bone interface in breast cancer [49]. The inhibition of TGFβ signaling by neutralizing antibodies or TGFBR1 inhibitors reduces osteolysis, reverses bone loss, and increases bone volume and mineral-to-collagen ratio in syngeneic models of metastatic breast cancer [49,50,51]. Reduced metastatic burden, osteoclast number and osteolysis upon TGFβ neutralization and TGFBR1 inhibition were also observed in mice injected with human MDA-MB-231 cell sublines preferentially metastasizing to the bone and lung [50]. Temporal analysis in mice indicates that blockade of TGFβ is less effective against established metastases than in early lesions [52]. Furthermore, the injection of MDA-MB-231 cells transfected with a dominant-negative mutant of TGFBR2 or the administration of oncolytic virus overexpressing soluble TGFBR2 receptor to sequester TGFβ results in less pronounced bone damage, the weaker expansion of osteoclasts and better survival [53,54]. Additionally, TGFBR2 on the surface of myeloid cells contributes to metastatic bone disease since myeloid-specific deletion TGFBR2 resulted in fewer bone lesions and lower bone tumor burden in athymic mice [55]. TGFβ exerts its prometastatic functions in metastatic bone disease in breast cancer at least partially through its canonical intracellular mediators, SMADs. The genetic depletion of SMAD4 in human and mouse breast cancer cells reduces the formation of osteolytic bone metastases and prolongs metastasis-free survival in mice [56,57]. Interestingly, two other key secondary messengers of the TGFβ axis, SMAD2 and SMAD3, appear to exert opposite functions in bone metastasis, with SMAD2 delaying the establishment and growth of bone metastases and SMAD3 driving an aggressive phenotype [58]. This effect can be partially explained by crosstalk between the TGFβ and other pathways. TGFβ has been shown to induce the expression of VEGF and CCL2 and increase microvessel density in breast cancer bone lesions in mouse models [58,59]. Basic fibroblast growth factor (bFGF), PTHrP, and cyclooxygenase 2 are further downstream effectors upregulated by TGFβ and involved in bone metastasis [53,60,61,62]. TGFβ also mediates bone loss induced by chemotherapy in breast cancer. It was shown that doxorubicin therapy leads to bone damage, the acceleration of bone metastasis growth, and the release of TGFβ and breast cancer cells to the circulation [63,64,65]. Furthermore, doxorubicin activates TGFβ signaling in breast cancer cells, which, in turn, promotes EMT and generation of the stem cell phenotype [64,65]. These stem cells show reduced sensitivity to doxorubicin, which could explain the development of resistance to chemotherapy in breast cancer patients. Importantly, TGFβ inhibition restores sensitivity to doxorubicin in animal models of breast cancer, decreases the incidence of bone metastases, reduces bone tumor burden, and increases osteoblasts mineralization and bone volume [64,65]. TGFβ blockade in prostate cancer models in mice has been shown by several studies to attenuate metastatic bone disease. For example, TGFBR1 inhibitor LY2109761 reduced the proliferation of osteolytic (PC-3) and osteoblastic (MDA-PCa-2b) human prostate cancer cells, reduced bone loss and turnover volume, prevented activation of osteoclasts, and increased bone mineral content and bone mineral density [66]. Additionally, the knockdown or inhibition of TGFβ in human PacMetUT1 cells decreased bone tumor growth and the incidence of bone metastases and reduced osteoblastic bone lesions [67]. Importantly, TGFβ blockade can not only prevent but also reduce already established metastatic bone disease. Oncolytic adenoviruses overexpressing soluble TGFBR2 were shown to decrease tumor growth, hypercalcemia, osteoclast number, and bone destruction in already established bone lesions [68]. Using cell-specific TGFBR2 knock-out mice, Meng et al. demonstrated that TGFβ signaling in osteoblasts inhibits bone metastases, while an opposite effect was observed in osteoclasts. The anti-metastatic properties of TGFβ in osteoblasts depend on the down-regulation of bFGF since tumor growth in osteoblast-specific TGFBR2 knock-out mice can be reversed by bFGF-neutralizing antibody [69]. Moreover, human bone-derived metastatic renal cell carcinoma cells overexpressing dominant-negative TGFBR1 and TGFBR2 are less capable of growing in the bone of athymic nude mice and display reduced osteolysis [70].

## 4. Conclusions

Spine metastases lead to significant morbidity, reduction in quality of life and worse prognosis. Considering that the spine is one of the most common locations of metastases in several types of cancer, the involvement of the spine is a significant health problem. To improve patient functioning and quality of life, vertebral metastases are frequently managed by surgery, radiotherapy or systemic therapy. Bisphosphonates are currently the standard of care systemic therapy aiming to reduce bone loss; however, their use is associated with considerable side effects and rather limited efficacy. Therefore, new treatments are urgently needed to effectively manage metastatic disease in the spine.

This systematic review is the first to summarize the available evidence on the functional role of cytokines and their receptors in the pathogenesis of spine metastases in solid tumors. Although during the review of records, we identified 68 cytokines/cytokine receptors to mediate bone metastases in 12 types of cancer, only 9 cytokines/cytokine receptors in four types of cancer appeared to play a role in metastases to the spine. This certainly does not indicate that spine metastasis is driven by only these nine cytokines/cytokine receptors. Instead, secondary cancer in the spine seems not to be specifically investigated or reported on by research studies, which is puzzling given the unmet clinical need caused by the morbidity and mortality associated with spine metastases. The fact that more cytokines/cytokine receptors have been established to play a functional role in bone metastases compared to the spine is a solid foundation for validation studies focused on the spine. An alternative explanation for the striking disproportion in the numbers of cytokines mediating bone and spine metastases could be that our search strategy was unable to identify all relevant records. For example, our broad search query did not return two clinical trials demonstrating a reduced incidence of new spine metastases in breast or prostate cancer patients treated with denosumab [16,17]. Nevertheless, the low number of trials reporting on the impact of therapies targeting cytokines on spine metastases is surprising, given their strong impact on patient prognosis.

In conclusion, the functional role of cytokines in spine metastases remains underexplored, and most of our understanding of this process appears to be derived from studies on other parts of the skeleton. The number of cytokines confirmed to mediate spinal metastasis is low compared with a vast spectrum of cytokines demonstrated to participate in the formation of secondary tumors in other parts of the skeleton. Additionally, it is unknown whether cytokines in the primary tumor could predetermine the homing of tumor cells to the spine. Therefore, further research is needed, including validation of the role of cytokines mediating metastases to other bones, in order to precisely address the unmet clinical need associated with spine metastases.

## Figures and Tables

**Figure 1 ijms-24-03785-f001:**
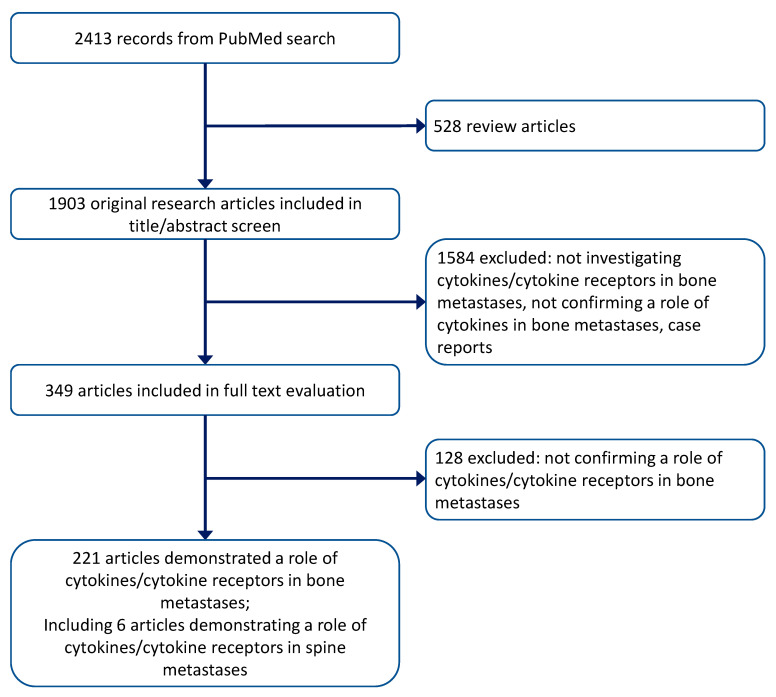
Preferred Reporting Items for Systematic Reviews and Meta-Analyses (PRISMA) flow chart of search criteria.

**Figure 2 ijms-24-03785-f002:**
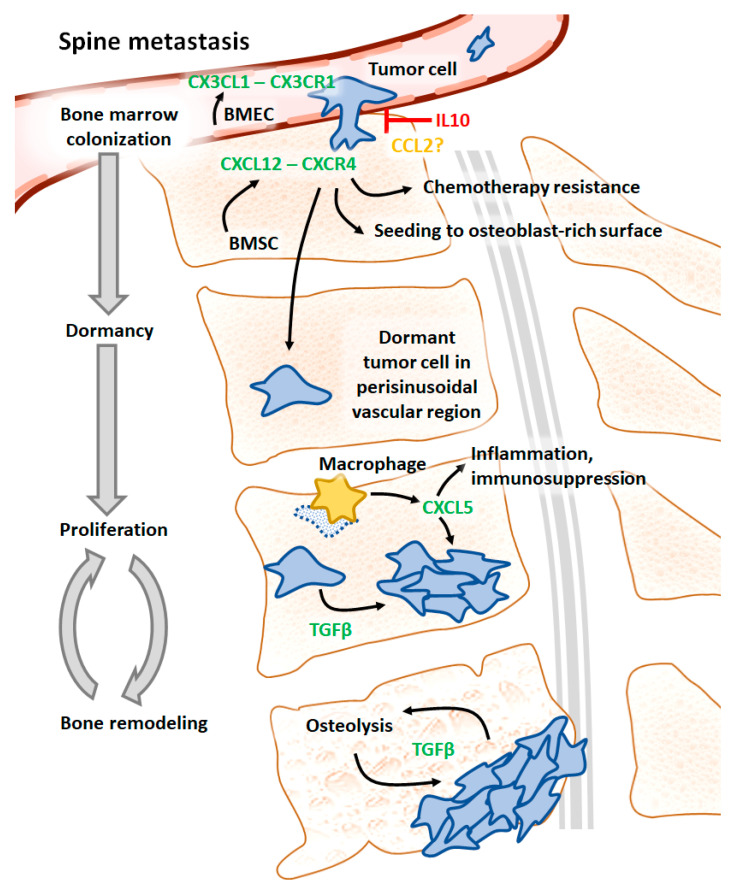
Involvement of cytokines across the stages of spine metastasis development. BMEC, bone marrow endothelial cells, BMSC, bone marrow stromal cells, CCL, CC motif chemokine ligand, CX3CL, CX3C motif chemokine ligand, CX3CR, CX3C motif chemokine receptor, CXCL, CXC motif chemokine ligand, CXCR, CXC motif chemokine receptor, IL, interleukin, TGF, transforming growth factor.

**Figure 3 ijms-24-03785-f003:**
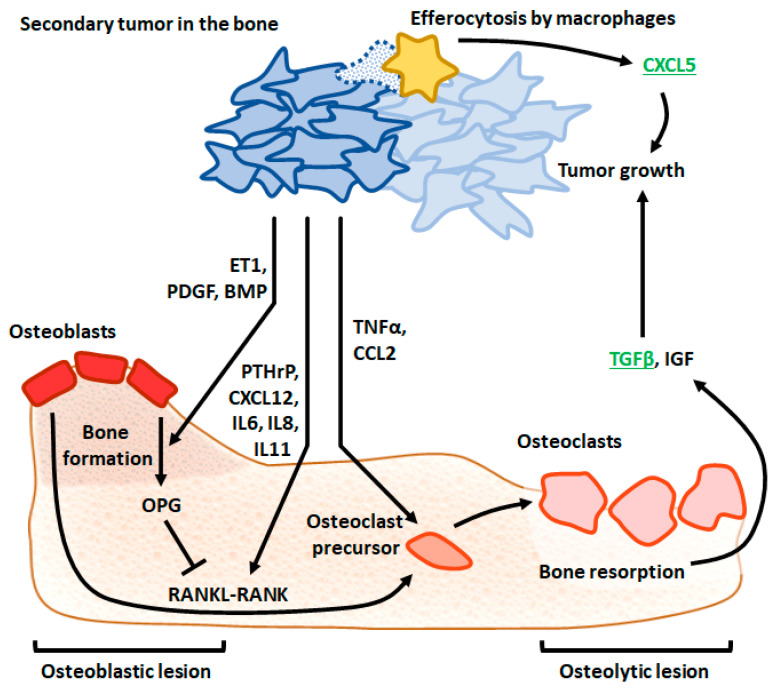
The vicious cycle of bone remodeling and tumor growth in bone metastasis [45,46]; cytokines confirmed to mediate this mechanism in the spine (CCL5 and TGFβ) are indicated in green. BMP, bone morphogenetic protein, CCL, CC motif chemokine ligand, CXCL, CXC motif chemokine ligand, ET, endothelin, IGF, insulin-like growth factor, IL, interleukin, OPG, osteoprotegerin, PDGF, platelet-derived growth factor, PTHrP, parathyroid hormone-related protein, RANK, receptor activator of nuclear factor-κB, RANKL, receptor activator of nuclear factor-κB ligand, TGF, transforming growth factor, TNF, tumor necrosis factor.

**Table 1 ijms-24-03785-t001:** List of identified articles demonstrating a functional link between cytokines and spine metastases.

Reference	Title	Cytokine, Cytokine Receptor	Primary Tumor	Setting
Sun et al. (2020), [19]	ADAM17-regulated CX3CL1 expression produced by bone marrow endothelial cells promotes spinal metastasis from hepatocellular carcinoma.	CX3CL1, CX3CR1	Liver (human hepatocellular carcinoma cell line MHCC97H)	Mice injected with CX3CR1-overexpressing MHCC97H cancer cells or with CX3CL1-silenced bone marrow endothelial cells
Roca et al. (2018), [20]	Apoptosis-induced CXCL5 accelerates inflammation and growth of prostate tumor metastases in bone.	CXCL5	Prostate (syngeneic murine prostate carcinoma cell line RM1)	Vertebral bodies inoculated with RM1-iC9 cells, subcutaneously implanted into CXCL5^−/−^ mice, followed by apoptosis induction using the FKBP fusion protein systems
Jung et al. (2013), [21]	Recruitment of mesenchymal stem cells into prostate tumours promotes metastasis.	CXCL12, CXCR4, CXCR6	Prostate (syngeneic murine prostate carcinoma cell line RM1)	RM1 cells treated with CXCL12 and/or CXCR4 inhibitor AMD3100, injected into CXCR6^−/−^ mice
Takahashi et al. (2009), [22]	Chemokine CCL2/MCP-1 negatively regulates metastasis in a highly bone marrow-metastatic mouse breast cancer model.	CCL2	Breast (murine mammary carcinoma cell line 4T1E)	Mice injected with CCL2-overexpressing or CCL2-silenced 4T1E cells
Bandyopadhyay et al. (2006), [23]	Inhibition of pulmonary and skeletal metastasis by a transforming growth factor-beta type I receptor kinase inhibitor.	TGFβ	Skin (human melanoma cell line MDA-MB-435, formerly described as of breast cancer origin)	TGFβ treatment in mice injected with MDA-MB-435-F-L cells
Stearns and Wang (1998), [24]	Antimestatic and antitumor activities of interleukin 10 in transfected human prostate PC-3 ML clones: Orthotopic growth in severe combined immunodeficient mice.	IL10	Prostate (human prostatic adenocarcinoma cell line PC-3)	IL10-overexpressing or IL10-treated PC-3 cells injected into mice

CCL, CC motif chemokine ligand, CX3CL, CX3C motif chemokine ligand, CX3CR, CX3C motif chemokine receptor, CXCL, CXC motif chemokine ligand, CXCR, CXC motif chemokine receptor, IL, interleukin, TGF, transforming growth factor.

## Data Availability

Not applicable.

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
