# Peer review of "The Role of Cytokines in the Metastasis of Solid Tumors to the Spine: Systematic Review"

_ijms, 2023, doi:10.3390/ijms24043785_

Round 1

Reviewer 1 Report

The review by Łabędź et. al., titled The role of cytokines in the metastasis of solid tumors to the 2 spine: systematic review discusses the role and involvement of cytokines in spine
metastasis, which is one of the complications frequently observed in patients with advanced carcinomas. The authors have discussed in details the role of CX3CL1, CX3CR1, IL10,
CCL2,CXCR6, CXCR4, CXCL12, CXCL5 & TGF-b in different cancer types. The review is extensive and relevant for readers.

My comments are as follows:

1. In section 2.3, line 148 There is evidence that CCL2 has opposite effects in prostate cancer and breast cancer, as shown by reported articles. Can the authors hypothesize
why? Since these disease types are gender specific, are there any reports of hormonal influence on CCL2 expression in patients?

2. The review seems wordy. I suggest adding more figures instead to show mechanism of action of the cytokines. Also, the key points/therapeutic options may be tabulated.

3. While the methodology of article selection as shown in Figure 1 is good, some articles may have been missed and should find mention in the review - e.g. Pompo et. al.,
Front Cell Dev Bio 2021 (https://doi.org/10.3389/fcell.2021.678532).

4. Minor edit Line 69 : Please correct the type for Bone marrow.

Author Response

The review by Łabędź et. al., titled “The role of cytokines in the metastasis of solid tumors to the 2 spine: systematic review” discusses the role and involvement of cytokines in spine metastasis, which is one of the complications frequently observed in patients with advanced carcinomas. The authors have discussed in details the role of CX3CL1, CX3CR1, IL10, CCL2,CXCR6, CXCR4, CXCL12, CXCL5 & TGF-b in different cancer types. The review is extensive and relevant for readers.

My comments are as follows:

  1. In section 2.3, line 148 – There is evidence that CCL2 has opposite effects in prostate cancer and breast cancer, as shown by reported articles. Can the authors hypothesize

why? Since these disease types are gender specific, are there any reports of hormonal influence on CCL2 expression in patients?

Response: We thank the Reviewer for this comment. When searching for the information on the gender-specific impact on CCL2 expression, we realized that the majority of evidence actually points towards the tumor promoting function of this chemokine, also in breast cancer. Therefore, we decided to rewrite this section to be more in line with the totality of available evidence.

  1. The review seems wordy. I suggest adding more figures instead to show mechanism of action of the cytokines. Also, the key points/therapeutic options may be tabulated.

Response: Following the Reviewer’s comment, we have substantially reduced the amount of text. We have also modified the existing figure for better overview on the function of cytokines specifically in the spine metastases and added another figure with more detailed visualization of mechanisms underlying the tumor growth and bone remodeling.

  1. While the methodology of article selection as shown in Figure 1 is good, some articles may have been missed and should find mention in the review - e.g. Pompo et. al.,

Front Cell Dev Bio 2021 (https://doi.org/10.3389/fcell.2021.678532).

Response: We are fully aware that our methodology was unable to capture all evidence on the role of cytokines in spine metastases, and therefore we mentioned this as a limitation of our article. We focused on searching for information on a direct mechanistic link, i.e. an evidence showing that a manipulation of cytokine levels or function had an impact on the development or persistence of spine metastases in ex vivo or in vitro. The publication mentioned by the Reviewer provides an insight into the interplay between cytokines and osteoblast by a comprehensive set of in vitro experiments followed by analysis of circulating markers for osteolytic marker TRACP5b and IL-8. Although these experiments are surely expanding our knowledge on the mechanisms at play during bone remodeling in cancer, they do not provide a direct functional link between cytokines and spine metastasis. For this reason, we refrained from citing the mentioned article to be in line with the methodology for identifying the relevant records.

  1. Minor edit – Line 69 : Please correct the type for “Bone” marrow.

Response: The typo has been corrected, thank you for pointing out.

Reviewer 2 Report

Review is written OK, however I encourage the authors to include figures and models pertaining to each section because else it is difficult for readers to understand the point of view.

I encourage the authors to emphasize the importance of spinal metastasis by including relevant statistics in the introduction, because the emphasis on disease burden is missing.

A comparative model of how cytokines impact spinal metastasis as compared other bone metastasis would be useful.

Author Response

Review is written OK, however I encourage the authors to include figures and models pertaining to each section because else it is difficult for readers to understand the point of view.

Response: We thank the Reviewer for this suggestion. We have revised the existing figure for better overview on the function of cytokines specifically in the spine metastases (i.e. those confirmed to directly play the role in this process). We also added another figure with more detailed visualization of mechanisms underlying the tumor growth and bone remodeling.

I encourage the authors to emphasize the importance of spinal metastasis by including relevant statistics in the introduction, because the emphasis on disease burden is missing.

Response: Following the Reviewer’s request, we have added a few more statistics and other information to highlight the burden of spine metastases.

A comparative model of how cytokines impact spinal metastasis as compared other bone metastasis would be useful.

Response: We have added a figure depicting the involvement of cytokines in tumor growth and bone tissue remodeling in bone metastases in general and highlighted those cytokines which have been proven to directly play a role in spine metastases. 

Reviewer 3 Report

1. Authors summarise data about the effect of cytokines on the metastasation of solid cancer to the spine. The majority of factors are chemokines.
2. Topic of review is not original but it improves knowledge about this highly hot topic of oncology.
3. In comparison with another review it brings new information freshly published in original articles. It is comprehensive and accompanied by one cartoon.
4. Concerning the improvement of quality I suggest reflecting on the fact that the majority of mentioned cytokines are chemokines and their receptors in the title of the article. Because of the importance of osteoclasts are crucial for bone metastasation I suggest including one paragraph related to the role of this cell.
5. Conclusion is consistent and reflects the main ideas of the article.
6. References are appropriate and up to date.
7. Include a schematic presentation summarising the effect of cytokines on the osteoclast activity during the bone metastazation.

Author Response

  1. Authors summarise data about the effect of cytokines on the metastasation of solid cancer to the spine. The majority of factors are chemokines.
    2. Topic of review is not original but it improves knowledge about this highly hot topic of oncology.
    3. In comparison with another review it brings new information freshly published in original articles. It is comprehensive and accompanied by one cartoon.
    4. Concerning the improvement of quality I suggest reflecting on the fact that the majority of mentioned cytokines are chemokines and their receptors in the title of the article. Because of the importance of osteoclasts are crucial for bone metastasation I suggest including one paragraph related to the role of this cell.
    Response: We thank the Reviewer for this comment. The Reviewer is absolutely correct in pointing out that the majority of cytokines identified to play the role in spine metastases are chemokines. However, changing the title to reflect this fact could signal the readers that our work was concentrated on analyzing the link between the chemokines (specifically) and spine metastases. Instead, our aim was more generally focused on cytokines and this approach allowed us to identify a vast amount of data on non-chemokine members of cytokine family, including IL10 and particularly growth factor TGFβ. Therefore, we feel that a more general title not pointing to a specific group of cytokines better reflects identified evidence. Nevertheless, we highlighted in the abstract the fact that majority of identified cytokines were chemokines. Regarding the importance of osteoclasts in bone metastases, we provided more information on the role of this cell type in the introduction and prepared additional figure on the cytokine-mediated involvement of osteoclasts in metastatic bone tissue remodeling. 

5. Conclusion is consistent and reflects the main ideas of the article.
6. References are appropriate and up to date.
7. Include a schematic presentation summarising the effect of cytokines on the osteoclast activity during the bone metastazation.
Response: As mentioned above, we have created a figure depicting the role of cytokines in the regulation of osteoclast-driven metastatic bone remodeling.